# Layer-Specific Damage Modeling of Porcine Large Intestine under Biaxial Tension

**DOI:** 10.3390/bioengineering9100528

**Published:** 2022-10-06

**Authors:** Aroj Bhattarai, Charlotte Anabell May, Manfred Staat, Wojciech Kowalczyk, Thanh Ngoc Tran

**Affiliations:** 1Department of Orthopaedic Surgery, University of Saarland, 66424 Homburg, Germany; 2Institute of Bioengineering, FH Aachen University of Applied Sciences, 52428 Jülich, Germany; 3Chair of Mechanics and Robotics, University of Duisburg-Essen, 47057 Duisburg, Germany

**Keywords:** biaxial tensile experiment, anisotropy, hyperelastic, constitutive modeling, damage

## Abstract

The mechanical behavior of the large intestine beyond the ultimate stress has never been investigated. Stretching beyond the ultimate stress may drastically impair the tissue microstructure, which consequently weakens its healthy state functions of absorption, temporary storage, and transportation for defecation. Due to closely similar microstructure and function with humans, biaxial tensile experiments on the porcine large intestine have been performed in this study. In this paper, we report hyperelastic characterization of the large intestine based on experiments in 102 specimens. We also report the theoretical analysis of the experimental results, including an exponential damage evolution function. The fracture energies and the threshold stresses are set as damage material parameters for the longitudinal muscular, the circumferential muscular and the submucosal collagenous layers. A biaxial tensile simulation of a linear brick element has been performed to validate the applicability of the estimated material parameters. The model successfully simulates the biomechanical response of the large intestine under physiological and non-physiological loads.

## 1. Introduction

The primary functions of the large intestine are to digest resistant starch, fiber and undigested proteins, to produce and absorb vitamins, electrolytes, short-chain fatty acids and amino acids, to absorb water, and to transport fecal matters outside the body [1,2]. Extending from the cecum to the anus, the intestinal wall is composed of four layers: outer serosa, muscle, submucosa and inner mucosa [3]. Each layer is a heterogeneous mixture of extracellular matrix and collagen fibers with two additional unidirectional smooth muscle fiber families in the longitudinal and the circumferential muscular layers [4,5]. Such arrangement offers a complex mechanical behavior characterized by a highly nonlinear anisotropic stress–strain relation, hysteresis and rate sensitivity, which have been widely demonstrated in uniaxial tensile, biaxial tensile and inflation experiments [3]. There is a large variability in the experiments and the constitutive models for the characterization of the intestines’ mechanical behavior [3], which are logically relevant for approximating its stress–strain relations. However, all the approaches are limited to the lower elastic range or only to the ultimate stress level. Furthermore, Egorov et al. in 2002 [6] observed two maxima in the human colorectal tissue tested in longitudinal direction, postulating the rupture of different layers at different stretch levels. The first higher narrow maximum at around 40% strain is hypothesized due to muscular rupture, and the second lower broad maximum after 80% strain due to the submucosal rupture. For almost two decades, such unique tissue mechanics have been completely neglected in the available characterization approaches, if they have ever been encountered at all. Most importantly, a correct selection of the constitutive model that is correlated with the experimental methodology, and capable of fully characterizing the material properties of such anisotropic soft tissues are necessary [7].

Intestines are subjected to progressive damage by remodeling during diseased condition or by staples/sutures after surgical anastomosis [8]. For example, ulcerative colitis is characterized by depleted mucosal lining, disrupted-thinned muscular collagen fibers [9], thickened submucosa and increased cross-sectional area [10,11]. Such microscopic changes result in different tissue mechanics than the healthier counterparts [10,11], which affects the normal physiological function of the intestine. Following the concept of effective stress reduction factors initially proposed by Kachanov [12], several deterministic, probabilistic and microstructural models have been proposed to describe the damage induced mechanical behavior of fiber-reinforced soft biological tissues [13]. Some of the recent structure-based damage evolution formulations for ligament, rectal sheath and mostly artery have been proposed by Balzani et al. [14], Calvo et al. [15], Volokh et al. [16], Peña et al. [17], Ehret & Itskov [18], Martins et al. [19], Comellas et al. [20], and Noble et al. [21]. The common assumption in all the formulations is that the global damage affects only the isochoric/volume-preserving or deviatoric/distortional part of the deformation as proposed by Simo [22], which can be additively decomposed into contributions from the isotropic ground matrix material and the anisotropic collagen fibers.

Research on intestine biomechanics is aimed to better understand and model anisotropy, viscoelasticity and heterogeneity [4,6,9,23,24,25,26,27,28,29,30,31,32,33,34,35]. To the best knowledge of the authors, this paper is the first to investigate mechanically-induced damage to the large intestine. Although a phase-field approach to model the fracture in the small intestine wall has been recently presented by Nagaraja et al. [36], the study is limited to uniaxial tension experiments, non-hysteresis mechanics, and overall damage parameters rather than a layer-or-fiber specific fracture analysis. The continuum damage evolution proposed by Comellas et al. [20] has been adapted in this paper to constitutively describe the true stress–strain response of the porcine large intestine during biaxial tensile loading until complete failure. Knowing the fact that mucosal and serosal collagen fibers do not contribute to the in-plane stiffness significantly [6,37], they have been therefore ignored in this study. Therefore, the overall tissue stiffness is provided by the extracellular matrix, the submucosal collagen, the longitudinal and the circumferential muscular collagen fiber families. For this, the fresh porcine intestine specimens have been biaxially stretched. The simple neo-Hookean and the well-known Holzapfel-Gasser-Ogden (HGO) constitutive model [38] have been used to describe the isotropic ground matrix and different collagen fiber families, respectively. The eventual damages on the load paths are defined by separate contributions of damage from each load-bearing collagen fiber family.

## 2. Materials and Methods

### 2.1. Equibiaxial Tensile Specimen Preparation

The large intestines of four pigs were obtained from the Institute of Laboratory Animal Science at the University Hospital RWTH Aachen, Germany. The experimental pig was primarily used for the medical training purposes and euthanized and dissected. Thus, the obtained intestine as a whole organ (Figure 1a) is covered with cloth soaked in a 0.9% NaCl solution to prevent the tissue from desiccation. Within an hour, the lump of a fresh intestine is brought to the testing facility (Biomechanics Laboratory, FH Aachen University of Applied Sciences, Jülich, Germany), divided into segments based on five anatomical coils, and completely cleaned of feces and other connective/adipose tissue. Each coil is taken in turn from which individual specimens with an edge length (*L*) of 40mm×40mm are prepared and the remaining tissue is stored in the 0.9% NaCl solution until the next testing. Close inspections have been done during preparation and the specimens with any preliminary damage are excluded for the biaxial tensile experiment. Specimen thickness was measured with a micrometer screw prior to testing.

### 2.2. Test Protocol

The equibiaxial tensile tests of the intestine specimens are carried out on the self-developed biaxial machine “BiAiX” (FH Aachen, Aachen, Germany, patent pending, DE 10 2017 116 067 A1) (Figure 1b). The BiAiX machine has already been used for several experiments, and the results have been published in a peer-reviewed journal [35]. It comprises of four arms with force measuring actuators and five light-weight aluminium cylindrical bars with a sharp needle at their free end for tissue fixation. The needles can move laterally to allow free stretching of each side of the specimen. For 2D planar evaluation, one Allied Vision Prosilica GT2450, 5 Megapixel camera (Allied Vision Technologies GmbH, Statroda, Germany) with a GigE Vision Gigabit Ethernet interface and a maximum frame rate of 15 f/s with AQUAMARINE 2.0/28 C lens (Schneider Kreuznach, Bad Kreuznach, Germany) lens is mounted above the specimen position in the BiAiX machine. The two force sensors of the BiAiX tension test machine are calibrated in an industrial manner with calibration weights. The camera is positioned perpendicular to the specimen surface. This guarantees an easy calibration for the 2D digital image correlation (DIC) with a calibration target. The specimens are then mounted in the apparatus ensuring the orientation of the sample always the same: longitudinal direction along the L-L axis and circumferential direction along the C-C axis (Figure 1b). To ensure a uniform evaluation of the specimens, the inner mucosal layer of the specimens is always oriented facing up and a graphite pattern (black random spots in Figure 1c,d) is sprayed for DIC. Polarized light with a polarizing filter in front of the camera lens is used to avoid reflections on the wet specimens. The measurement protocols for the tensile test are as follows: five cycles of preconditioning followed by a monotonous stretching both at the speed of 10 mm/min until failure and a frame rate of 1 Hz. Figure 1, panels c and d, show specimen images in the undeformed and highly stretched configurations, respectively with fixed camera-lens position.

### 2.3. Multilayer Passive Anisotropic Strain Energy Function

Recent advancement in the histological techniques such as nonlinear second harmonic generation (SHG) has greatly improved the architectural description of the multilayered intestine [5,37,39]. Maier et al. [5] have found the orientation and thickness of the collagen fibers in the submucosa layers to be *⌀* 6.5 μm, aligned in a crosswise arrangement at $\alpha =\pm 30^{\circ }$ to the longitudinal direction (Figure 2). Moreover, collagen fibers (*⌀* 1.9 μm) are also present in the longitudinal and the circumferential muscular layers. The collagen fiber orientations in different layers can be defined by the angles measured between the fiber families and the Cartesian coordinate axis ($e_{1}$) as:(1)$$\begin{array}{c} a_{01}=\left[\begin{array}{c} \cos0^{\circ }\\ \sin0^{\circ }\\ 0 \end{array}\right]=\left[\begin{array}{c} 1\\ 0\\ 0 \end{array}\right];a_{02}=\left[\begin{array}{c} \cos90^{\circ }\\ \sin90^{\circ }\\ 0 \end{array}\right]=\left[\begin{array}{c} 0\\ 1\\ 0 \end{array}\right];\\ a_{03}=\left[\begin{array}{c} \cos30^{\circ }\\ \sin30^{\circ }\\ 0 \end{array}\right]=\left[\begin{array}{c} \frac{\sqrt{3}}{2}\\ \frac{1}{2}\\ 0 \end{array}\right];a_{04}=\left[\begin{array}{c} \cos(-30^{\circ })\\ \sin(-30^{\circ })\\ 0 \end{array}\right]=\left[\begin{array}{c} \frac{\sqrt{3}}{2}\\ -\frac{1}{2}\\ 0 \end{array}\right], \end{array}$$
where the collagen fibers in the longitudinal and the circumferential muscular layers are perpendicular to each other (Figure 2).

Within the framework of nonlinear elasticity, such fiber-reinforced composites can be mechanically described by highly deformable hyperelastic constitutive model expressed in terms of the quasi-isochoric (negligible volume change under loading) strain energy functions (SEF). The decoupled representation of SEF adopted from Holzapfel et al. [38]
(2)$$\begin{array}{c} \psi \left(J,\bar{C}\right):=\psi_{\mathrm{vol}}^{\circ }\left(J\right)+{\bar{\psi }}_{\mathrm{isch}}\left({\bar{I}}_{1},a_{01},a_{02},a_{03},a_{04}\right), \end{array}$$
is based on the kinematic multiplicative decomposition of the deformation gradient tensor, $F=F^{\circ }\bar{F}$ into a spherical or volumetric ($F^{\circ }=J^{\frac{1}{3}}I$) and a unimodular or isochoric part ($\bar{F}=J^{-\frac{1}{3}}F$), [40]. J:=det[F] is the determinant of the deformation gradient tensor, a measure of the volume change. It allows the definition of the isochoric right and left Cauchy–Green deformation tensors as: $\bar{C}={\bar{F}}^{T}\bar{F}=J^{-\frac{2}{3}}C$ and $\bar{B}=\bar{F}{\bar{F}}^{T}=J^{-\frac{2}{3}}B$, respectively. In Equation (Equation 2),
(3)$$\begin{array}{c} \psi_{\mathrm{vol}}^{\circ }\left(J\right)=\frac{\kappa_{0}}{2}{\left(J-1\right)}^{2}, \end{array}$$
is the purely volumetric contribution of the SEF and $\kappa_{0}$ is the initial bulk modulus. The purely isochoric term ${\bar{\psi }}_{\mathrm{isch}}$ can be additively split into the isotropic part (${\bar{\psi }}_{\mathrm{iso}}$) from the matrix and the anisotropic part (${\bar{\psi }}_{\mathrm{aniso}}$) from the collagen fibers. The isochoric SEF in Equation (Equation 2) can be expressed as:(4)$${\bar{\psi }}_{\mathrm{isch}}\left({\bar{I}}_{1},a_{01},a_{02},a_{03},a_{04}\right)={\bar{\psi }}_{\mathrm{iso}}\left({\bar{I}}_{1}\right)+{\bar{\psi }}_{\mathrm{aniso}}\left(a_{01},a_{02},a_{03},a_{04}\right)={\bar{\psi }}_{\mathrm{iso}}\left({\bar{I}}_{1}\right)+{\bar{\psi }}_{\mathrm{aniso}}\left({\bar{I}}_{4,a_{01}},{\bar{I}}_{4,a_{02}},{\bar{I}}_{4,a_{03}},{\bar{I}}_{4,a_{04}}\right),$$
where ${\bar{I}}_{1}:=\mathrm{tr}\left[\bar{C}\right]$ is the modified first principal invariant of the symmetric modified right Cauchy–Green tensor and ${\bar{I}}_{4,a_{0i}}:=\bar{C}:A_{0i}=\bar{C}:\left(a_{0i}⊗a_{0i}\right)=a_{0i}\cdot \bar{C}\cdot a_{0i}=\lambda_{i}^{2}$ represent the anisotropic invariant, which is quantitatively equal to the square of the fiber stretches $\lambda_{a_{0i}}^{2}$ associated with the $i^{\mathrm{th}}$ collagen fiber family, $a_{0i}$. The deformation gradient F transforms the reference fiber direction into current configuration, $a_{i}=F\left[a_{0i}\right]$ and the respective structural tensors, $A_{i}:=FA_{0i}F^{T}$.

At low strain, the tissue mechanical behavior is solely due to the ground matrix and can be defined using the classical neo-Hookean material model:(5)$$\begin{array}{c} {\bar{\psi }}_{\mathrm{iso}}\left({\bar{I}}_{1}\right)=\frac{\mu_{0}}{2}\left({\bar{I}}_{1}-3\right), \end{array}$$
where $\mu_{0}>0$ is the initial shear modulus and has the unit of stress. Siri et al. [37] measured similar mechanical behavior of the submucosa and the muscular layers. Therefore, each layer in this study has been modeled with the same form of the HGO constitutive model, represented by an exponential function [38]:(6)$${\bar{\psi }}_{\mathrm{aniso}}\left({\bar{I}}_{4,a_{01}},{\bar{I}}_{4,a_{02}},{\bar{I}}_{4,a_{03}},{\bar{I}}_{4,a_{04}}\right)=\left\{\begin{array}{cc} \sum_{i=1}^{4}\frac{k_{1,a_{0i}}}{2k_{2,a_{0i}}}\left[\exp\left\{k_{2,a_{0i}}{\left({\bar{I}}_{4,a_{0i}}-1\right)}^{2}\right\}-1\right]\; & \mathrm{if}\;{\bar{I}}_{4a_{0i}}\geq 1\\ 0 & \mathrm{otherwise}, \end{array}\right.$$
where $k_{1,a_{0i}}>0$ are stress-like material parameters and $k_{2,a_{0i}}$ are dimensionless parameters for the $i^{\mathrm{th}}$ collagen fiber family and influence the intestine mechanics at high strain only resulting in a polyconvex SEF. For more detail, refer to Bhattarai et al. [35]. Considering the measurement of the muscular (mus) layer collagen fibers (∅ 1.9 μm) by Feng et al. [39], the longitudinal and the circumferential muscle layer both can be assumed to be transversely isotropic, governed by the common material parameters, $k_{1,\mathrm{mus}}$ and $k_{2,\mathrm{mus}}$. Likewise, the submucosal (sm) collagen fibers (∅ 6.5 μm) arranged in a crosswise alignment and assuming no fiber interaction contribute to the second part of the anisotropic stiffness for which $k_{1,\mathrm{sm}}$ and $k_{2,\mathrm{sm}}$ are the material parameters. Finally, the anisotropic HGO constitutive model for the intestine (Equation (Equation 6)) can be re-written as:(7)$$\begin{array}{c} \begin{array}{c} {\bar{\psi }}_{\mathrm{aniso}}={\bar{\psi }}_{\mathrm{mus}}\left({\bar{I}}_{4,a_{01}},{\bar{I}}_{4,a_{02}}\right)+{\bar{\psi }}_{\mathrm{sm}}\left({\bar{I}}_{4,a_{03}},{\bar{I}}_{4,a_{04}}\right),\\ \mathrm{where}\;{\bar{\psi }}_{\mathrm{mus}}\left({\bar{I}}_{4,a_{01}},{\bar{I}}_{4,a_{02}}\right)=\frac{k_{1,\mathrm{mus}}}{2k_{2,\mathrm{mus}}}\sum_{i=1}^{2}\left[\exp\left\{k_{2,\mathrm{mus}}{\left({\bar{I}}_{4,a_{0i}}-1\right)}^{2}\right\}-1\right]\\ {\bar{\psi }}_{\mathrm{sm}}\left({\bar{I}}_{4,a_{03}},{\bar{I}}_{4,a_{04}}\right)=\frac{k_{1,\mathrm{sm}}}{2k_{2,\mathrm{sm}}}\sum_{j=3}^{4}\left[\exp\left\{k_{2,\mathrm{sm}}{\left({\bar{I}}_{4,a_{0j}}-1\right)}^{2}\right\}-1\right], \end{array} \end{array}$$
satisfying the condition ${\bar{I}}_{4a_{0i}}$ and ${\bar{I}}_{4a_{0j}}\geq 1$. Adding Equations (3), (5) and (7), the original intestinal SEF Equation (Equation 2) becomes:(8)$$\psi =\frac{\kappa_{0}}{2}{\left(J-1\right)}^{2}+\frac{\mu_{0}}{2}\left({\bar{I}}_{1}-3\right)+\frac{k_{1,\mathrm{mus}}}{2k_{2,\mathrm{mus}}}\sum_{i=1}^{2}\left[\exp\left\{k_{2,\mathrm{mus}}{\left({\bar{I}}_{4,a_{0i}}-1\right)}^{2}\right\}-1\right]+\frac{k_{1,\mathrm{sm}}}{2k_{2,\mathrm{sm}}}\sum_{j=3}^{4}\left[\exp\left\{k_{2,\mathrm{sm}}{\left({\bar{I}}_{4,a_{0j}}-1\right)}^{2}\right\}-1\right].$$

### 2.4. Anisotropic Hyperelastic Stress Calculation

The mechanical description of the intestine wall requires the derivation of an appropriate stress as a function of strain. The hyperelastic response of the intestinal tissue defined by the SEF in Equation (Equation 8) can be written in terms of the second Piola–Kirchhoff stress (S) with respect to invariants [22]. Using the chain rule, the Cauchy stress (σ) can be computed as: (9)$$\begin{array}{cc} \sigma & :=\frac{1}{J}FSF^{T}=\frac{1}{J}F\left(2\frac{\partial \psi }{\partial C}\right)F^{T}=\frac{1}{J}F\left(2\frac{\partial \psi_{\mathrm{vol}}^{\circ }}{\partial J}\frac{\partial J}{\partial C}+2\frac{\partial {\bar{\psi }}_{\mathrm{iso}}}{\partial {\bar{I}}_{1}}\frac{\partial {\bar{I}}_{1}}{\partial I_{1}}\frac{\partial I_{1}}{\partial C}+2\sum_{i=1}^{4}\frac{\partial {\bar{\psi }}_{\mathrm{aniso}}}{\partial {\bar{I}}_{4,a_{0i}}}\frac{\partial {\bar{I}}_{4,a_{0i}}}{\partial I_{4,a_{0i}}}\frac{\partial I_{4,a_{0i}}}{\partial C}\right)F^{T}\\ \; & =\frac{1}{J}F\left(\underset{S_{\mathrm{vol}}^{\circ }}{\underbrace{pJC^{-1}}}\right)F^{T}+\frac{1}{J}F\left(\underset{{\bar{S}}_{\mathrm{iso}}}{\underbrace{\mu \left(J^{-\frac{2}{3}}I-\frac{1}{3}{\bar{I}}_{1}C^{-1}\right)}}\right)F^{T}+\frac{1}{J}F\left(\underset{{\bar{S}}_{\mathrm{mus}}}{\underbrace{2\sum_{i=1}^{2}{\bar{\psi }}_{\mathrm{mus}}^{\prime }\left({\bar{I}}_{4,a_{0i}}\right)J^{-\frac{2}{3}}\mathrm{DEV}\left(A_{0i}\right)}}\right)F^{T}\\ \; & +\frac{1}{J}F\left(\underset{{\bar{S}}_{\mathrm{sm}}}{\underbrace{2\sum_{i=3}^{4}{\bar{\psi }}_{\mathrm{sm}}^{\prime }\left({\bar{I}}_{4,a_{0i}}\right)J^{-\frac{2}{3}}\mathrm{DEV}\left(A_{0i}\right)}}\right)F^{T}\\ \; & =\underset{\sigma_{\mathrm{vol}}^{\circ }}{\underbrace{pI}}+\underset{{\bar{\sigma }}_{\mathrm{iso}}}{\underbrace{\frac{\mu }{J}\left(\bar{B}-\frac{1}{3}{\bar{I}}_{1}I\right)}}+\underset{{\bar{\sigma }}_{\mathrm{mus}}}{\underbrace{\frac{2}{J}{\bar{\psi }}_{\mathrm{mus}}^{\prime }\left({\bar{I}}_{4,a_{01}}\right)\mathrm{dev}\left(A_{1}\right)+\frac{2}{J}{\bar{\psi }}_{\mathrm{mus}}^{\prime }\left({\bar{I}}_{4,a_{02}}\right)\mathrm{dev}\left(A_{2}\right)}}+\underset{{\bar{\sigma }}_{\mathrm{sm}}}{\underbrace{\frac{2}{J}{\bar{\psi }}_{\mathrm{sm}}^{\prime }\left({\bar{I}}_{4,a_{03}}\right)\mathrm{dev}\left(A_{3}\right)+\frac{2}{J}{\bar{\psi }}_{\mathrm{sm}}^{\prime }\left({\bar{I}}_{4,a_{4}}\right)\mathrm{dev}\left(A_{4}\right)}}, \end{array}$$
where $p=\kappa_{0}\left(J-1\right)$ is the hydrostatic pressure; I is the second order identity tensor, $\mathrm{DEV}\left(A_{0i}\right)=A_{0i}-\frac{1}{3}\left(A_{0i}:C\right)C^{-1}$ is the material deviator of the structural tensor $A_{0i}$ with the right Cauchy–Green tensor **C** operating as metric tensor and $\mathrm{dev}\left(A_{i}\right)=A_{i}-\frac{1}{3}\left(A_{i}:I\right)I$ is the deviator of the structural tensor $A_{i}$ associated with the current fiber orientation $a_{i}=F\left[a_{0i}\right]$. The derivative of the $j^{\mathrm{th}}$ anisotropic SEF component related to the muscular and submucosal collagen fibers can be computed as:(10)$$\begin{array}{c} {\bar{\psi }}_{j}^{\prime }\left({\bar{I}}_{4,a_{0i}}\right)=\frac{\partial {\bar{\psi }}_{j}}{\partial {\bar{I}}_{4,a_{0i}}}=k_{1,a_{0i}}\left({\bar{I}}_{4,a_{0i}}-1\right)\exp\left\{k_{2,a_{0i}}{\left({\bar{I}}_{4,a_{0i}}-1\right)}^{2}\right\}\;\forall i=1,2,3,4;\;j=\mathrm{mus},\mathrm{sm}. \end{array}$$

### 2.5. Nonlinear Damage Constitutive Model

In this section, the basic formulation of the damage evolution developed by Comellas et al. [20] has been adopted and presented for the investigation of the damage in the intestine material. Damage of any material is characterized by the reduction of the stiffness at stretch beyond the physiological range, due to microstructural failures, followed by an eventual fracture of the specimen. Initially postulated by Simo [22], damage phenomena are assumed to affect only the isochoric parts of the constitutive Equation (Equation 8). However, the damage in each of the components has to be distinguished in order to provide a complete description of the intestine damage. Weisbecker et al. [41] performed layer-specific damage experiments, and used a pseudo-elastic damage model to find aorta damage primarily induced by the collagen fibers. Furthermore, the non-collagenous matrix can be completely damaged well below the physiological loading range. Hence, only the collagenous layer damage is considered in this paper as:(11)$$\begin{array}{c} \psi =\psi_{\mathrm{vol}}^{\circ }\left(J\right)+{\bar{\psi }}_{\mathrm{iso}}\left({\bar{I}}_{1}\right)+\left(1-D_{\mathrm{lm}}\right){\bar{\psi }}_{\mathrm{mus}}\left({\bar{I}}_{4,a_{01}}\right)+\left(1-D_{\mathrm{cm}}\right){\bar{\psi }}_{\mathrm{mus}}\left({\bar{I}}_{4,a_{02}}\right)+\left(1-D_{\mathrm{sm}}\right){\bar{\psi }}_{\mathrm{sm}}\left({\bar{I}}_{4,a_{03}},{\bar{I}}_{4,a_{04}}\right), \end{array}$$
where ${\bar{\psi }}_{\mathrm{iso}},{\bar{\psi }}_{\mathrm{mus}},{\bar{\psi }}_{\mathrm{sm}}$ are the undamaged isochoric SEF for isotropic matrix, muscular and submucosal layers, respectively. The Kachanov-like reduction factors $\left(1-D_{\mathrm{lm}}\right),\left(1-D_{\mathrm{cm}}\right)$ and $\left(1-D_{\mathrm{sm}}\right)$ are the functions of the normalized scalars with $D_{k}\in \left[0,1\right]$ defined as internal damage variables, where k=lm for longitudinal muscular layer collagen fibers, cm for circumferential muscular layer collagen fibers, and sm for submucosal collagen fibers.

Utilizing the concept of Clausius–Duhem inequality for isothermal cases, the internal dissipation can be generalized as
(12)$$\begin{array}{c} D_{\mathrm{int}}=\frac{1}{2}S:\dot{C}-\dot{\psi }\geq 0, \end{array}$$
where the rate of the SEF in Equation (Equation 11) using the chain rule can be written as
(13)$$\begin{array}{c} \dot{\psi }=\frac{\partial \psi_{\mathrm{vol}}^{\circ }\left(J\right)}{\partial C}:\dot{C}+\frac{\partial {\bar{\psi }}_{\mathrm{iso}}\left({\bar{I}}_{1}\right)}{\partial C}:\dot{C}+\left(1-D_{\mathrm{lm}}\right)\left(\frac{\partial {\bar{\psi }}_{\mathrm{mus}}\left({\bar{I}}_{4,a_{01}}\right)}{\partial C}:\dot{C}\right)+\left(1-D_{\mathrm{cm}}\right)\left(\frac{\partial {\bar{\psi }}_{\mathrm{mus}}\left({\bar{I}}_{4,a_{02}}\right)}{\partial C}:\dot{C}\right)\\ +\left(1-D_{\mathrm{sm}}\right)\left(\frac{\partial {\bar{\psi }}_{\mathrm{sm}}\left({\bar{I}}_{4,a_{03}},{\bar{I}}_{4,a_{04}}\right)}{\partial C}:\dot{C}\right)-{\dot{D}}_{\mathrm{lm}}{\bar{\psi }}_{\mathrm{mus}}\left({\bar{I}}_{4,a_{01}}\right)-{\dot{D}}_{\mathrm{cm}}{\bar{\psi }}_{\mathrm{mus}}\left({\bar{I}}_{4,a_{02}}\right)-{\dot{D}}_{\mathrm{sm}}{\bar{\psi }}_{\mathrm{sm}}\left({\bar{I}}_{4,a_{03}},{\bar{I}}_{4,a_{04}}\right). \end{array}$$

Comparing the expressions in Equations (Equation 12) and (Equation 13), the Kachanov effective second Piola–Kirchhoff stress for finite strain and the non-negative internal dissipation can be deduced as
(14)$$\begin{array}{cc} \; & S=2\frac{\partial \psi_{\mathrm{vol}}^{\circ }\left(J\right)}{\partial C}+2\frac{\partial {\bar{\psi }}_{\mathrm{iso}}\left({\bar{I}}_{1}\right)}{\partial C}+2\left(1-D_{\mathrm{lm}}\right)\frac{\partial {\bar{\psi }}_{\mathrm{mus}}\left({\bar{I}}_{4,a_{01}}\right)}{\partial C}+2\left(1-D_{\mathrm{cm}}\right)\frac{\partial {\bar{\psi }}_{\mathrm{mus}}\left({\bar{I}}_{4,a_{02}}\right)}{\partial C}+2\left(1-D_{\mathrm{sm}}\right)\frac{\partial {\bar{\psi }}_{\mathrm{sm}}\left({\bar{I}}_{4,a_{03}},{\bar{I}}_{4,a_{04}}\right)}{\partial C},\\ \; & \mathrm{and},\;D_{\mathrm{int}}={\dot{D}}_{\mathrm{lm}}{\bar{\psi }}_{\mathrm{mus}}\left({\bar{I}}_{4,a_{01}}\right)+{\dot{D}}_{\mathrm{cm}}{\bar{\psi }}_{\mathrm{mus}}\left({\bar{I}}_{4,a_{02}}\right)+{\dot{D}}_{\mathrm{sm}}{\bar{\psi }}_{\mathrm{sm}}\left({\bar{I}}_{4,a_{03}},{\bar{I}}_{4,a_{04}}\right)\geq 0, \end{array}$$
respectively. Inequality (Equation 14) shows that damage is a dissipative process which implies non-decreasing damage variables, i.e., $D_{k}\in \left[0,1\right]\;\&\;{\dot{D}}_{k}\geq 0\;\forall k=\left\{\mathrm{lm},\;\mathrm{cm},\;\mathrm{sm}\right\}$. Applying the push forward, the Kachanov effective Cauchy stress tensor can be obtained as
(15)$$\begin{array}{c} \sigma =\sigma_{\mathrm{vol}}^{\circ }\left(J\right)+{\bar{\sigma }}_{\mathrm{iso}}\left({\bar{I}}_{1}\right)+\left(1-D_{\mathrm{lm}}\right){\bar{\sigma }}_{\mathrm{mus}}\left({\bar{I}}_{4,a_{01}}\right)+\left(1-D_{\mathrm{cm}}\right){\bar{\sigma }}_{\mathrm{mus}}\left({\bar{I}}_{4,a_{02}}\right)+\left(1-D_{\mathrm{sm}}\right){\bar{\sigma }}_{\mathrm{sm}}\left({\bar{I}}_{4,a_{03}},{\bar{I}}_{4,a_{04}}\right). \end{array}$$

### 2.6. Damage Evolution

At any current loading time *t*, Simo [22] proposed $\phi_{t,k}=G\left(\tau_{t,k}\right)-G\left(\tau_{t,k}^{\max}\right)\leq 0$ as the damage criterion in the strain space with the condition $\phi_{t,k}=0$ defining the damage surface. Here, $\tau_{t,k}=\sqrt{2{\bar{\psi }}_{k}\left(\bar{C}\left(t\right)\right)}$ is the equivalent strain as the norm at current time *t* related to the undamaged deviatoric SEF, ${\bar{\psi }}_{k}\left(\bar{C}\left(t\right)\right)$ and $\tau_{t,k}^{\max}=\max_{s=0,t}\sqrt{2{\bar{\psi }}_{k}\left(\bar{C}\left(s\right)\right)}$ is the maximum value up to current time *t* or the current damage threshold which satisfies $\tau_{t,k}^{\max}\geq \tau_{0,k}^{d}$, with $\tau_{0,k}^{d}$ as the initial damage threshold stress. Damage is initiated, if the current $\tau_{t,k}\geq \tau_{t,k}^{\max}$ which is equal to the initial damage threshold stress at the beginning of the deformation, i.e., at $t=0,\tau_{0,k}^{\max}=\tau_{0,k}^{d}$.

To complete the constitutive model with failure, the exponential evolution formulation used by Comellas et al. [20] is adapted in this paper: the evolution of the damage variables is given as
(16)$$\begin{array}{c} D_{k}=G\left(\tau_{k}\right)=1-\frac{\tau_{0,k}^{d}}{\tau_{k}}\exp\left[A_{k}\left(1-\frac{\tau_{k}}{\tau_{0,k}^{d}}\right)\right];\;A_{k}={\left[\frac{g_{f,k}^{d}}{{\left(\tau_{0,k}^{d}\right)}^{2}}-\frac{1}{2}\right]}^{-1}\forall \;k=\mathrm{lm},\;\mathrm{cm},\;\mathrm{sm}. \end{array}$$
where $\tau_{0,k}^{d}$ is the initial damage threshold stress; $g_{f,k}^{d}$ is the fracture energy per unit volume, and are the material parameters to be determined through fitting the experiment data. The damage variables for each component are defined for the interval $D_{t,k}\in \left[0,1\right]$ that satisfies $D_{0,k}=\left(\tau_{0,k}\right)=0$ and $D_{\infty ,k}=G\left(\tau_{\infty ,k}\right)=1$.

### 2.7. Numerical Simulations

Numerical simulation has been a cost-effective technique to validate the feasibility of the adopted mathematical formulations and the reliability of the estimated parameters that describe the mechanical behavior of any experimented material. In this study, the finite element (FE) analysis is performed in the commercial FE software LS-DYNA (Ansys, Inc., Canonsburg, PA, USA) Version: smp d R11.1.0 in 64 bit Windows 10. A user-defined Fortran material subroutine that takes into account a) the hyperelastic response of the four collagen fibers, and b) the damage evolution in the load bearing collagen fibers (Equation (Equation 15)) has been coded inside LS-DYNA. The first part of the subroutine implementation has been successfully tested in Bhattarai et al. [35]. An example is presented in this paper to demonstrate the numerical performance of the damaged induced anisotropic hyperelastic model and the estimated material parameters. A cube of edge length 1 mm meshed with one linear brick element and reinforced by four fiber families (Figure 3) is considered. The four collagen fiber families are represented by unit directional vectors, $a_{01},a_{02},a_{03},a_{04}$. For biaxial tension in the XY plane, four nodes in the bottom surface are constrained to move along the *z*-axis, while equibiaxial strain is controlled by displacements, which are prescribed on all nodes in the XY plane.

## 3. Results

### 3.1. Equibiaxial Tensile Experiments

The mean specimen thickness varied from 0.68 to 2.15 mm, with a mean of 1.188 mm (N = 102). Displacement-controlled (10 mm/min) equibiaxial extension has been applied on the intestine specimens, which took 12 min on average to completely rupture each specimen. The tensile force ($P_{L}$ and $P_{C}$) are recorded at each load step on actuators of the longitudinal and the circumferential directions spanning over 3.3–17.0 N and 3.8–19.55 N, respectively.

The experimental methodology reported in this paper endures a technical problem, which limits the number of valid tests. As shown in Figure 4, the tissue tearing at the fixation with increasing stretch is merely inevitable due to early global fracture at the needle insertion points causing a sudden contraction of the specimen in the opposite of the loading direction. This phenomenon is no longer continuum damage and does not provide information about the intended damage in the tissue. In contrast to the expected stress-softening, unloading like behavior, i.e., backward and downward path of force–displacement or stress–strain curves occurs, which is purely due to tissue tearing at the holes making such data invalid for the continuum damage analysis. All the tested specimens in this study that showed global fracture at the needle insertions were therefore excluded from the damage analysis.

For the DIC evaluation with the ISTRA 4D (Limess Messtechnik und Software GmbH, Krefeld, Germany) software, residuum = 40–50, facet size > 50 and grid spacing = 1/3 facet size > 17 are used to obtain the homogeneous Green–Lagrange strain field without a significant loss of the grid points. The Green–Lagrange strain ($E_{LL},E_{CC}$) in both loading directions are evaluated directly from the heat maps using the DIC technique in the ISTRA 4D software for each individual image and recorded as an averaged value from a chosen deformation region (Figure 4). It is interesting that inhomogenous and anisotropic strain distributions are observed. In specimens S1, S3, and S5, several in-plane ripples are observed that look similar to calm ocean waves directed vertically for the longitudinal load direction and horizontally for the circumferential direction. This is not clearly understood, but the underlying collagen fibers might have withstood the applied stress. A higher magnification and better resolution camera, if available, may be used to resolve such strain distribution.

### 3.2. Damage Based Anisotropic Intestine Mechanics

Using the incompressibility condition, the true or Cauchy stress ($\sigma_{LL}=P_{L}\lambda_{L}/LT$ and $\sigma_{CC}=P_{C}\lambda_{C}/LT$) are set as a function of the elongation, $\lambda_{L}=\sqrt{2E_{LL}+1}$ and $\lambda_{C}=\sqrt{2E_{CC}+1}$. Figure 5 shows the representative Cauchy stress–stretch curves of the intestines from four pigs in the longitudinal (blue curves) and in the circumferential (red curves) directions under equibiaxial tensile stretching. The porcine intestine showed clear anisotropy with stiffer mechanical response in the longitudinal direction than in the circumferential direction. The standard J-shaped stress–stretch curves are observed for all specimens followed by the reduction of the stress slope induced by microstructural failure of the intestine wall composure. Furthermore, different tissue failure stretches are observed in both directions. Therefore, different values of the damage variables are expected in each direction due to progressive damages of individual fibers during the course of the tensile stretching. Significantly lower (by approximately two orders of magnitude) shear strains than the normal strain are found and are therefore neglected in this study.

### 3.3. Damage Parameter Estimation

To determine the damage driven passive, incompressible, anisotropic, hyperelastic material parameters as per the SEF in Equation (Equation 15), the curve fittings are performed in Matlab (The MathWorks, Inc., Natick, MA, USA). A careful selection of the experimental curves is made: the non J-shaped and the specimens without concurrent softening stress stretch curves in both longitudinal and circumferential directions are excluded for the model parametrization. The coefficients of determination $R^{2}\in \left[0,1\right]$ in the longitudinal ($R_{L}^{2}$) and in the circumferential ($R_{C}^{2}$) directions, respectively, are used to determine the quality of the fit for the estimated material parameters, $R^{2}$ is calculated as follows:(17)$$\begin{array}{c} R^{2}=1-\frac{\sum_{i=1}^{n}{\left(\sigma_{i}^{\exp}-\sigma_{i}^{\mathrm{fit}}\right)}^{2}}{\sum_{i=1}^{n}{\left(\sigma_{i}^{\exp}-\sigma^{\mathrm{mean}}\right)}^{2}}. \end{array}$$
where *n* is the number of measured data points, $\sigma_{i}^{\exp}$ is the Cauchy stress measured experimentally, $\sigma_{i}^{\mathrm{fit}}$ is the corresponding stress values predicted by the fitting procedure using the SEF, and $\sigma^{\mathrm{mean}}$ is the mean of the experimental stress. A value of $R^{2}$ closer to unity indicates a good fit to the experimental data. The fitted stress–stretch curves are shown in Figure 5 and the material parameters are listed in Table 1. For each specimen, unique discontinuities or sudden changes in the stress–stretch slopes are associated with the damages in the longitudinal, the circumferential, and the submucosal collagen fibers governed by the initial threshold stresses and the fracture energies for each layer (Table 1). Figure 6 shows the predicted layer-specific damages for all the fitted specimens. In 50% (S1 and S3) of the studied specimens, the circumferential collagen may initiate damage followed by the submucosal collagen and the longitudinal collagen. However, the longitudinal layer shows a brittle-like failure at a lower stretch than the circumferential and the submucosal layers.

### 3.4. Numerical Simulation of the Biaxial Tension

Based on the user defined material subroutine, the biaxial tension of all pigs has been simulated using a linear brick element as shown in Figure 7. Using LS-DYNA with 16 CPUs, the total simulation time is only 2 min. Under prescribed elongations, DX = 0.31 mm and DY = 0.38 mm and using material parameters as given in Table 1, the brick model is contracted by DZ = −0.6493 mm along the *z*-axis (Figure 7). The Cauchy stress and the strain data in the linear element are extracted from LS-PrePost. The respective stretches are computed, and the Cauchy stress vs. stretch curves are plotted over the experiment-fit dataset (Figure 5). The details of the experimented large intestine mechanics, such as anisotropy, hyperelasticity and damage, have been appropriately generated, i.e., the material behavior for each specimen is qualitatively and quantitatively similar to the dataset. This confirms that the estimated material parameters are numerically stable, and the implemented Fortran user material subroutine is capable for further numerical investigations.

## 4. Discussion

### 4.1. Large Intestine Mechanical Behavior

Abundant resources on the large intestine biomechanics can be found to describe its mechanical behavior that are used for various computational applications [3,8,25,27,28,37,42,43,44,45]. However, these studies are limited to the uniaxial tensile tests, non-fresh experimental specimen, and undamaged passive anisotropic hyperelastic material modeling. For a complete and realistic material description, appropriate experimental techniques and computational models are necessary. Focused on such aspects, equibiaxial tensile experiments of fresh porcine colon, and the layer-specific orthotropic material description beyond the supraphysiological loading have been performed in this study. In general, the large intestine is found to be clearly orthotropic, with the standard J-shaped nonlinear stress–stretch paths under tensile loading (Figure 5). Furthermore, the stiffness in the longitudinal direction is found to be larger than the stiffness in the circumferential direction, with greater average failure stretch in the latter direction (1.37 vs. 1.52). Thus, the results obtained are consistent with the previous studies. For example, dynamic equibiaxial tension tests were performed on the human cruciate colon specimens by Howes and Hardy [27] and observed a greater average failure $2^{\mathrm{nd}}$ Piola–Kirchhoff stress (3.20±1.51 MPa) and a lower average Green–Lagrange failure strain (0.139±0.039) in the longitudinal direction, than in the circumferential direction (2.35±1.37 MPa) and (0.158±0.036). Likewise, Sokolis and co-authors performed inflation/extension tests on rat proximal colon and rectum specimens [4,45]. Each section of the intestine possessed greater longitudinal stiffness than the circumferential stiffness for which the Fung exponential [46] and the HGO constitutive models [38] were used. Similar differences between two directions have been obtained in biaxial tension by Siri et al. [32,37] and Puértolas et al. [34]. However, these studies are limited to the lower strain regime (≈15%) and may not provide complete information of the tissue mechanics.

Other aspects in the constitutive modeling of the fiber-reinforced anisotropic soft tissues that are not discussed in this study are in-plane fiber dispersion, interaction between two intertwined fibers, and the fiber shear contribution [47,48,49,50]. For any two intertwined fiber families, e.g., submucosal collagen fibers, $a_{03},a_{04}$, the anisotropic invariants ${\bar{I}}_{5}=a_{03}\cdot {\bar{C}}^{2}\cdot a_{03}$ and ${\bar{I}}_{7}=a_{04}\cdot {\bar{C}}^{2}\cdot a_{04}$, contributes to the fiber shear deformation, which have a strong correlation with $I_{4}$ and $I_{6}$. However, they may lead to an ill-posed parameter estimation problem and are usually not included in the constitutive formulation [51], whereas the invariant, $I_{8}=\left(a_{03}\cdot a_{04}\right)a_{03}\cdot \bar{C}a_{04}$ represents the interaction between the in-plane intertwined fiber families [52]. Likewise, an additional tissue stiffness can be provided taking into account the fiber dispersion within the tissue by modifying the fiber family function, Equation (Equation 7) based on the Generalized Structure Tensor approach as:(18)$${\bar{\psi }}_{\mathrm{sm}}\left({\bar{I}}_{4,a_{03}},{\bar{I}}_{4,a_{04}}\right)=\frac{k_{1,\mathrm{sm}}}{2k_{2,\mathrm{sm}}}\sum_{j=3}^{4}\left[\exp\left\{k_{2,\mathrm{sm}}\left(\left(1-\kappa \right)\left({\bar{I}}_{1}-3\right)+\kappa {\left({\bar{I}}_{4,a_{0j}}-1\right)}^{2}\right)\right\}-1\right],$$
where κ (not to confuse with $\kappa_{0}=\mathrm{initial}\;\mathrm{bulk}\;\mathrm{modulus}$) is a measure of dispersion in the fiber orientation. For κ=0, the model reduces to perfectly aligned fibers and the case associated with κ=1/3 is isotropically distributed fibers. For more information on the derivation and its application, see [47,48,53].

### 4.2. Damage Evolution of the Intestine Layers

Existing studies on the large intestine are limited to the undamaged passive anisotropy and do not include the damage evolution in the mathematical modeling. To the best of the authors’ knowledge, Nagaraja et al. [36] are the only work so far that models the fracture in the intestinal wall, but uses it for the small intestine. With a phase-field modeling approach, the failure of the rectangular notched strip is governed by the fracture toughness ($G_{C}>0$) with two additional anisotropy material parameters. Using the neo-Hookean and the four-fiber family based HGO strain energy function, the standard intestine anisotropy (stiffer longitudinal direction) has been well predicted. However, the uniaxial tensile testing and the non-layer-specific fracture parametrization do not fully differentiate the damage progressions in all load-bearing layers.

In order to fill this gap, we emphasize the characterization of the equibiaxially stretched large intestine based on the damage evolutions in all the load-bearing collagenous layers. Herein, the collagen fiber families in the longitudinal muscular layer, in the circumferential muscular layer, and in the submucosal layer contribute to the applied load. Therefore, damage ($D_{k},\forall k=\mathrm{lm},\;\mathrm{cm},\;\mathrm{sm}$) is calculated separately for these fiber families depending on their stiffness related material parameters ($\mu_{0},k_{1,i},k_{2,i},\forall i=\mathrm{mus},\;\mathrm{sm}$), initial damage threshold stress ($\tau_{0,k}^{d}$), and fracture energy ($g_{f,k}^{d}$). The influence of the damage related parameters is such that, the lower the value of threshold stress ($\tau_{0,k}^{d}$) for a particular layer, the earlier the damage initiates in that layer. Likewise, the lower the value of fracture energy ($g_{f,k}^{d}$), the steeper the negative slope of the damage stress–stretch curve becomes. From the curve fit using Equation (Equation 16) in Figure 6, the following sequences of the damages in the intestine layers have been predicted. The specimens S1 and S3 may show early circumferential layer damage, followed by the submucosal and the longitudinal muscular layer. In specimen S2 with the lowest longitudinal stiffness among all, the longitudinal layer damage is immediately followed by the submucosal layer and the circumferential layer. However, in specimen S4, the submucosal layer damage may have followed by the longitudinal and the circumferential. In all the specimens, the instantaneous (brittle-like) failure of the longitudinal muscular layer is typically observed right after the damage initiation. Furthermore, the circumferential layer and the submucosal layer failure show slower failure that progresses with increasing stretch. Although the experimental data-set fit well in all the presented specimens, damage sequence and evolution are not consistent. The biological variance in animal could be the reason why the specimens from each subject are unique. Therefore, a larger number of specimens is necessary from more animal subjects and is essential to conclude the exact event of the layer failure.

In modeling damage of the fiber-reinforced anisotropic material, localization occurs due to spurious mesh dependency of the softening constitutive model [54]. It means that, when failure initiates in a particular finite element, depending on the material properties and boundary conditions, the deformation increases rapidly and the element collapses. At the same deformation state, the neighboring elements may still be intact to resist more deformation. This generates rapid stress oscillations between such adjoining elements, and a homogeneously distributed stress cannot be achieved. Such pathological mesh dependent effects can be limited by the nonlocal strategy: introduction of an additional non-local variable in the constitutive behavior at a material point, which is dependent on the neighboring material point. Therefore, instead of using the local damage parameter, an integral average damage can be calculated within the interaction domain or the radius of influence, commonly called an internal length that regulates the size of the damaging zone. The non-local formulation based on Lemaitre’s model may greatly benefit simulating damage based impact and crash simulations. For more information, see [55,56,57].

### 4.3. Limitations of This Study

This study possesses numerous advantages such as appropriate biaxial tensile experiments, realistic supraphysiological loadings for damage evaluation, fiber-reinforced anisotropic mathematical modeling, and numerically stable material parameters validated by the finite element simulations. Each specimen has shown a strict repeatability and consistency in relation to maximum load, and higher stiffness in the longitudinal direction than in the circumferential direction. However, this study is still limited to only four representative experiments on four pig subjects, which is not statistically significant. Studies have shown that the mechanical behavior of the large intestine varies throughout its entire length from the proximal colon to the rectum. In this regard, the investigation direction should be further extended in the future to an adequately large number of pig subjects throughout its entire length from the proximal colon to the rectum for better statistical comparison and biological variance. Furthermore, one of the major difficulties in the biaxial tensile test using needles is the localized premature tearing, for which a better fixation technique would be helpful. Finally, in order to predict the layer-specific damage evolution, experiments on the separated intestinal wall would be more realistic. However, with our current experience, the separation has not been straightforward because the tissue was traumatized in the process before the experiment, rendering the specimen useless for the mechanical damage analysis. Therefore, a better technique is desired if the objective is to be achieved. Nevertheless, the numerical approach presented in this study is still relevant and the objective to present an appropriate numerical tool for the computational analysis of the damage in the multi-layered tubular architecture has been met.

## 5. Conclusions

Fiber-reinforced gastrointestinal biomechanics is not a very new subject of study, but damage has not yet been explored. Damage mechanics is important, especially for the lower bowel diseases and associated treatments. In such a situation, the load bearing collagen families across the thickness are greatly remodeled, altering the stiffness in both longitudinal and circumferential directions. Consequently, the load-stretch resisted prior symptoms will no longer be safe or damage-free. Likewise, for symptoms that cannot be easily treated with medication, anastomosis is commonly performed in which the diseased part is dissected, and two tissue sections are rejoined. On doing so, the tissue junction can be severely traumatized, both globally and locally. In order to achieve a successful treatment technique that is as safe as possible, an appropriate experimental methodology, and a realistic material description is necessary. With a focus on these aspects, the equibiaxial tensile experiment of the fresh porcine large intestine and the layer-specific anisotropic material description beyond the supraphysiological loading have been performed, and the results are presented in this study.

## Figures and Tables

**Figure 1 bioengineering-09-00528-f001:**
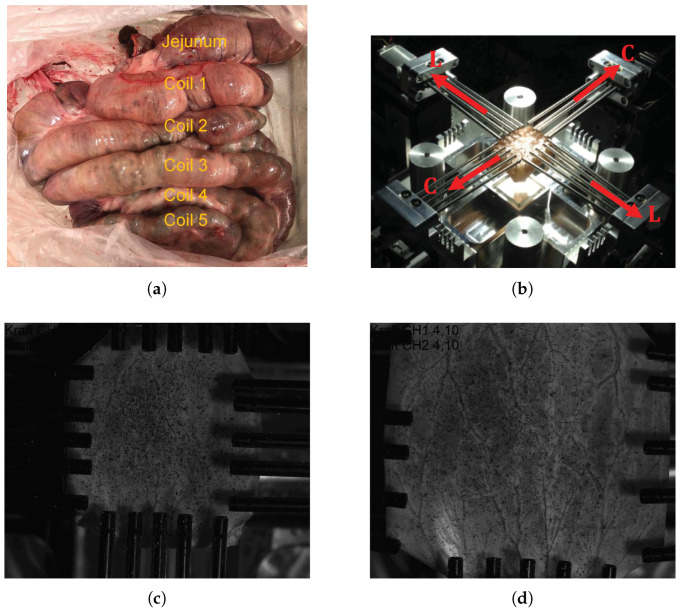
(**a**) Porcine large intestine as an intact organ prior to segmentation showing jejunum and five colon coils; (**b**) biaxial tensile test set up illuminated by the polarized light showing fiber orientations and the device coordinate system, and images of the test specimen in the (**c**) undeformed configuration and (**d**) deformed configuration taken from ISTRA4D.

**Figure 2 bioengineering-09-00528-f002:**
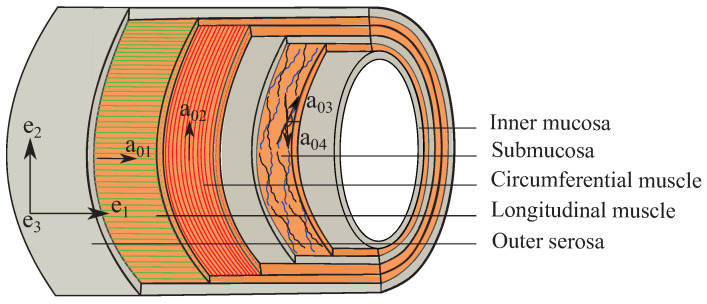
Schema of the multilayer colon showing collagen fiber orientations in the longitudinal muscular ($a_{01}$), the circumferential muscular ($a_{02}$) and the submucosal ($a_{03},a_{04}$) layers with respect to the Cartesian coordinate system ($e_{1},e_{2},e_{3}$).

**Figure 3 bioengineering-09-00528-f003:**
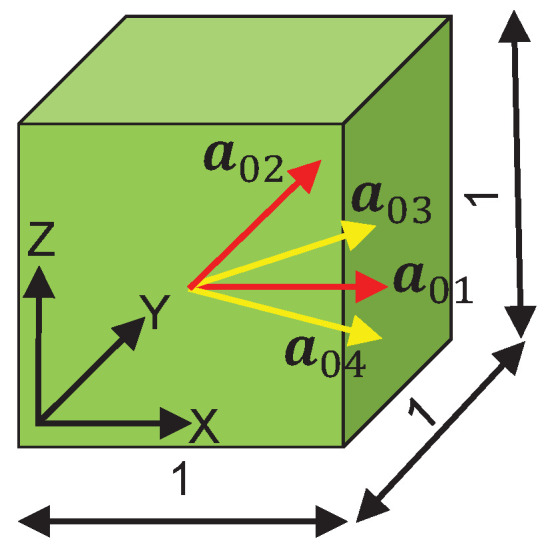
A linear brick FE showing fiber directions with dimension units in mm.

**Figure 4 bioengineering-09-00528-f004:**
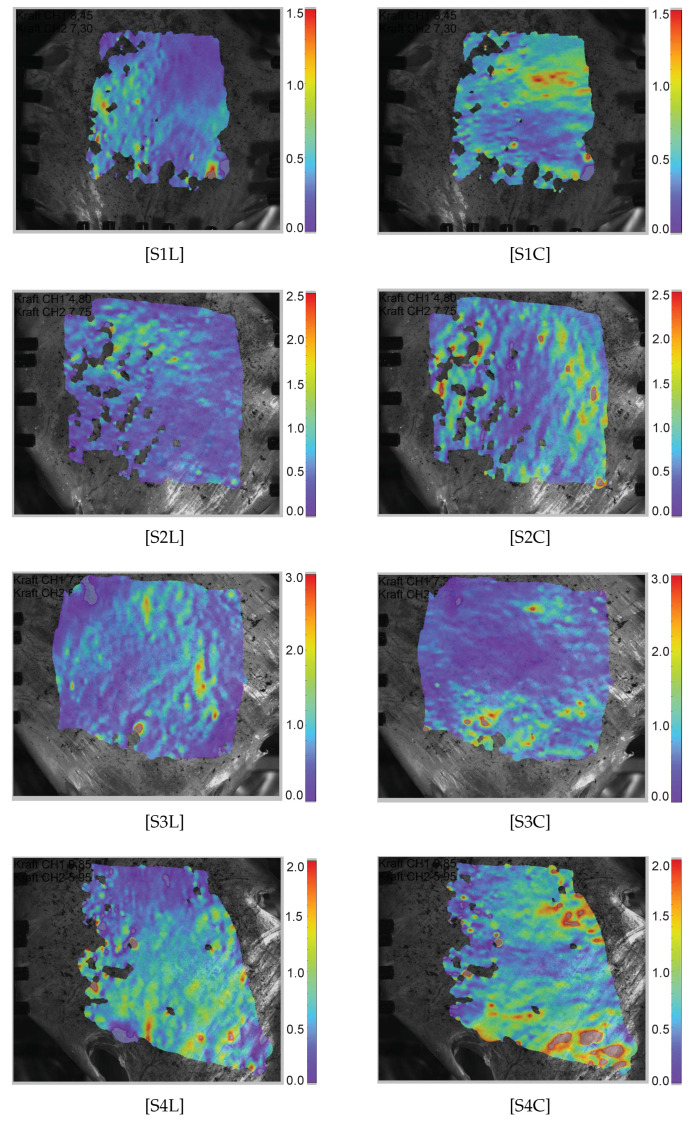
Green–Lagrange strain heat maps ($E_{LL},E_{CC}$) measured along the longitudinal and the circumferential direction in the specimens. “S*i*L” represents the $i^{\mathrm{th}}$ specimen along the longitudinal direction, whereas “S*i*C” represents the $i^{\mathrm{th}}$ specimen along the circumferential direction.

**Figure 5 bioengineering-09-00528-f005:**
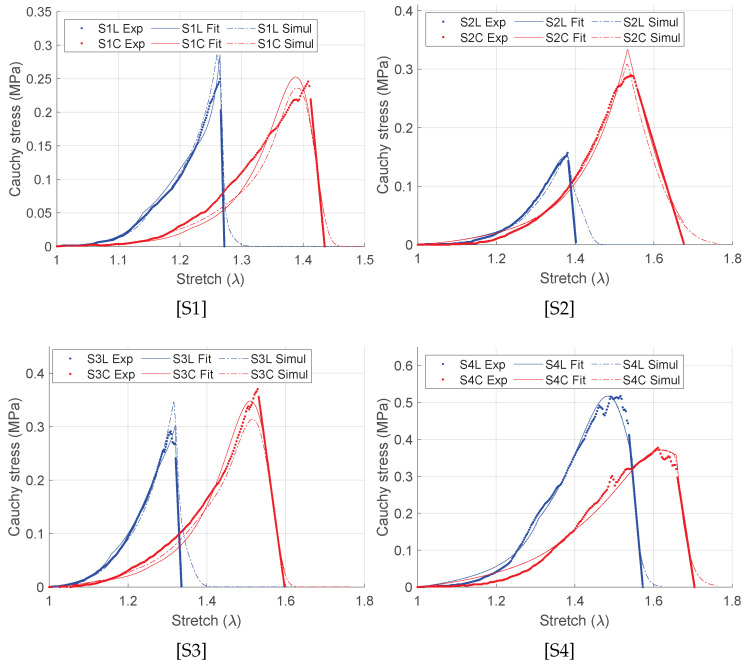
Fitting of the representative stress–stretch curves using damage formulation (Equation (Equation 17)). “Exp” represents the experiment data set, “Fit” represents the Matlab curve fit data set, and “Simul” represents the simulation values.

**Figure 6 bioengineering-09-00528-f006:**
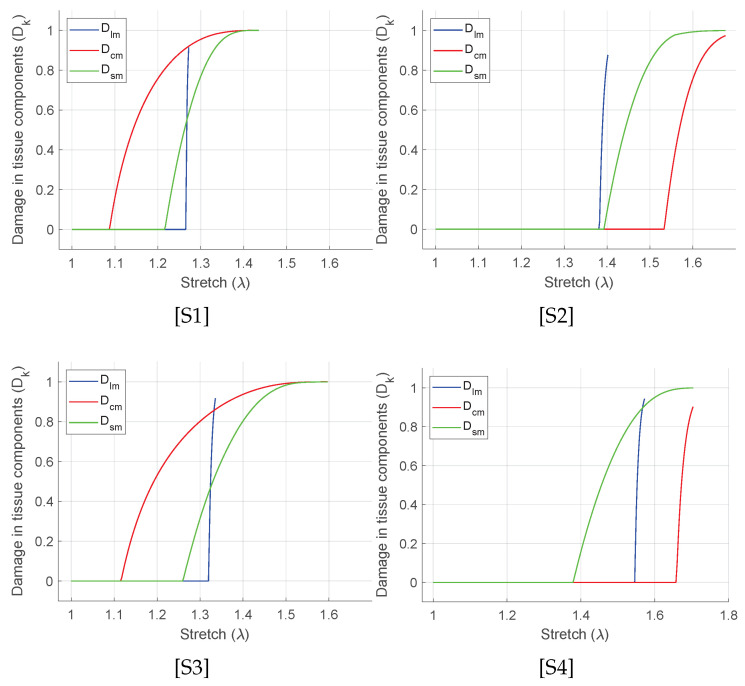
Evolution of damage in each layers $D_{\mathrm{lm}},D_{\mathrm{cm}}$ and $D_{\mathrm{sm}}$ for the curve fit as shown in Figure 5.

**Figure 7 bioengineering-09-00528-f007:**
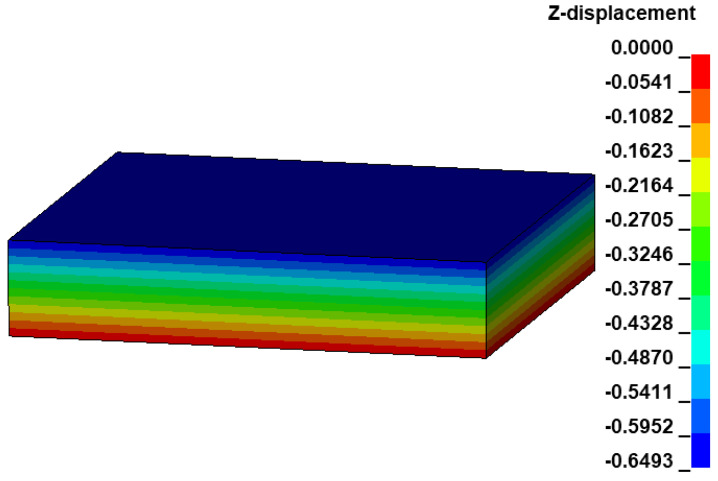
Deformation of the linear brick element under prescribed biaxial tension on the XY plane for the specimen S1.

**Table 1 bioengineering-09-00528-t001:** Fitted values of anisotropic HGO material parameters including damage.

No.	$\mu_{0}$	$k_{1,\mathrm{mus}}$	$k_{2,\mathrm{mus}}$	$k_{1,\mathrm{sm}}$	$k_{2,\mathrm{sm}}$	$\tau_{0,\mathrm{lm}}^{d}$	$g_{f,\mathrm{lm}}^{d}$	$\tau_{0,\mathrm{cm}}^{d}$	$g_{f,\mathrm{cm}}^{d}$	$\tau_{0,\mathrm{sm}}^{d}$	$g_{f,\mathrm{sm}}^{d}$	$R_{L}^{2}$	$R_{C}^{2}$
	(MPa)	(MPa)		(MPa)		(MPa)${}^{\frac{1}{2}}$	(MPa)	(MPa)${}^{\frac{1}{2}}$	(MPa)	(MPa)${}^{\frac{1}{2}}$	(MPa)		
S1	0.0001	0.0030	10.6781	0.0068	13.3155	0.1149	0.0073	0.0110	0.0169	0.0624	0.0125	0.9793	0.9377
S2	0.0001	0.0138	0.6804	0.0002	7.4843	0.1241	0.0083	0.2236	0.0387	0.0379	0.0089	0.9931	0.9852
S3	0.0001	0.0096	4.1802	0.0094	7.0591	0.1439	0.0118	0.0257	0.0346	0.0750	0.0224	0.9855	0.9824
S4	0.0001	0.0272	0.0144	0.0055	2.9810	0.2308	0.0278	0.2921	0.0461	0.1302	0.0738	0.9873	0.9775

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
