# Peer review of "Layer-Specific Damage Modeling of Porcine Large Intestine under Biaxial Tension"

_bioengineering, 2022, doi:10.3390/bioengineering9100528_

Round 1

Reviewer 1 Report

The authors aim to constitutively model the stress-strain response of the porcine large intestine during biaxial tensile loading until complete failure for use in describing the impact of damage induced by intestinal diseases and surgery.  To do this, they perform equibiaxial tensile experiments using fresh porcine colon (N = 4) to provide a layer-specific orthotropic material description beyond the supraphysiological loading.  They take into account the contribution of collagen “families” in the longitudinal muscular layer, the circumferential layer and the submucosal layer to mechanical behavior and failure.  Repeatedly find the specimens are anisotropic with higher stiffness in the longitudinal vs circumferential direction. Perform a validation study using experimentally estimated material parameters in a biaxial tensile simulation of a linear brick element and successfully simulate the mechanical behavior of the large intestine under physiological and non-physiological loads.  Strengths of the study include the need for such models that take into account damage behavior of a heterogeneous multi-layered tissue and the acquisition of experimental data for use in the constitutive model.  The main limitation of the study is the small sample size (N = 4 large pig intestine) which clearly could have been increased given the accessibility of porcine intestine and may have been able to overcome some of the problems with biologic variability.  Other less problematic limitations are the exclusion of samples throughout the study without clear justification or clarification of how many were excluded.  This would be important so as to limit the amount of bias.  Overall, the study is well written, the experimental design is adequately justified, and the model experimentally sound.

Author Response

The authors would like to thank the reviewer for their detailed remarks and suggestions for our paper. These comments have indeed helped a lot in improving the paper, and have been incorporated in the revised manuscript. All the changes made in the revised paper are in red text. Our point-to-point responses to the reviewers’ comments and the revisions made are detailed below:

We were aware that our study is limited to only 4 representative samples from 4 pigs. Unfortunately, the Covid 19 pandemic became a major setback in the expected progress of this research project (March 2020 – March 2022) in regards to the availability of the porcine specimens for medical training and laboratory experiment. The Institute of Laboratory Animal Science at the University Hospital RWTH Aachen, Germany had to limit their medical trainings into virtual platform, thus, the predicted number of animal dissection got drastically cut off. The tested sample size (N = 4) is indeed quantitatively small for the establishment of the statistical significance of the estimated data, which unwillingly has been the major limitation of this study. However, the experiments were repeatable and consistent with higher stiffness in the longitudinal direction than in the circumferential direction. Nevertheless, tests on large number of pig subjects should be considered in future for better statistical comparison and biological variance.

     We have stated this limitation in the manuscript. Furthermore, the following comments were also added in the section 3.1, between lines 242 - 252 of the revised manuscript.

     The experimental methodology reported in this paper endures a technical problem, which limits the number of valid tests. As shown in Fig. 4, the tissue tearing at the fixation with increasing stretch are merely inevitable due to early global fracture at the needle insertion points causing a sudden contraction of the specimen in opposite of the loading direction. This phenomenon is no longer continuum damage and does not provide information about the intended damage in the tissue. In contrast to the expected stress-softening, unloading like behavior, i.e., backward and downward path of force-displacement or stress-strain curves occurs, which is purely due to tissue tearing at the holes making such data invalid for the continuum damage analysis. All the tested specimens that showed global fracture at the needle insertions were therefore excluded from the damage analysis. 

Reviewer 2 Report

This article describes mechanical testing, a constitutive model, and finite element implementation of the model of the large intestine. Biaxial experiments were performed on porcine intestinal tissue, recording stretch ratio and stress data. The data was then modeled using a hyperelastic HGO model combined with damage modeling. The constitutive model was tested in Ls-dyna. The manuscript presents results that would potentially enable future research in the biomechanics of the intestine and related devices.

Author Response

Thanks for your comments

Reviewer 3 Report

This is an interesting work, well developed, that would benefit from some references that are missing mainly from Ogden and co-authors in different aspects and in particular on biaxial tests. The reference

Merodio J, Ogden R (eds) (2020) Constitutive modelling of solid continua, solid mechanics and its applications, vol 262. Springer, Berlin

deals with many aspects included in the paper and this reference

On planar biaxial tests for anisotropic nonlinearly elastic solids. A continuum mechanical framework

GA Holzapfel, RW Ogden Mathematics and Mechanics of Solids 14 (5), 474-489, 2009   is an important one dealing wih these aspects, to give a few references. More can be found by these authors in the subject easily.

Author Response

The authors would like to thank the reviewers for their detailed remarks and suggestions for our paper. These comments have indeed helped a lot in improving the paper, and have been incorporated in the revised manuscript. All the changes made in the revised paper are in red text. Our point-to-point responses to the reviewers’ comments and the revisions made are detailed in the attached file.
Response
We were aware that our study is limited to only 4 representative samples from 4 pigs. Unfortunately, the Covid 19 pandemic became a major setback in the expected progress of this research project (March 2020 – March 2022) in regards to the availability of the porcine specimens for medical training and laboratory experiment. The Institute of Laboratory Animal Science at the University Hospital RWTH Aachen, Germany had to limit their medical trainings into virtual platform, thus, the predicted number of animal dissection got drastically cut off. The tested sample size (N = 4) is indeed quantitatively small for the establishment of the statistical significance of the estimated data, which unwillingly has been the major limitation of this study. However, the experiments were repeatable and consistent with higher stiffness in the longitudinal direction than in the circumferential direction. Nevertheless, tests on large number of pig subjects should be considered in future for better statistical comparison and biological variance. 
     We have stated this limitation in the manuscript. Furthermore, the following comments were also added in the section 3.1, between lines 242 - 252 of the revised manuscript. 
     The experimental methodology reported in this paper endures a technical problem, which limits the number of valid tests. As shown in Fig. 4, the tissue tearing at the fixation with increasing stretch are merely inevitable due to early global fracture at the needle insertion points causing a sudden contraction of the specimen in opposite of the loading direction. This phenomenon is no longer continuum damage and does not provide information about the intended damage in the tissue. In contrast to the expected stress-softening, unloading like behavior, i.e., backward and downward path of force-displacement or stress-strain curves occurs, which is purely due to tissue tearing at the holes making such data invalid for the continuum damage analysis. All the tested specimens that showed global fracture at the needle insertions were therefore excluded from the damage analysis.   
